# A Rotary Compression Process for Producing Hollow Gear Shafts

**DOI:** 10.3390/ma13245718

**Published:** 2020-12-15

**Authors:** Arkadiusz Tofil, Janusz Tomczak, Tomasz Bulzak, Zbigniew Pater, Marcin Buczaj, Andrzej Sumorek

**Affiliations:** 1Faculty of Mechanical Engineering, Lublin University of Technology, 36 Nadbystrzycka Str, 20-618 Lublin, Poland; j.tomczak@pollub.pl (J.T.); t.bulzak@pollub.pl (T.B.); z.pater@pollub.pl (Z.P.); 2Faculty of Electrical Engineering and Computer Science, Lublin University of Technology, 38A Nadbystrzycka Str, 20-618 Lublin, Poland; m.buczaj@pollub.pl (M.B.); a.sumorek@pollub.pl (A.S.)

**Keywords:** rolling, rotary compression, hollow parts, gear shafts

## Abstract

This paper presents selected numerical and experimental results of a study investigating the process of forming hollow stepped gear shafts from tubes by rotary compression. The objective of the study was to determine whether the rotary compression process is an effective method of producing hollow stepped gear shafts and to identify limitations of this manufacturing method. A theoretical analysis involved the numerical modeling of the proposed process by the finite element method (FEM). 3D simulations were performed using the commercial simulation software package Simufact Forming. The analysis involved examining the material flow pattern along with thermal and force parameters of the process. The FEM results were verified with experimental tests conducted under laboratory conditions. The experiments were performed on a machine specially designed for the rotary compression of hollow parts. Results demonstrate that it is difficult to form a stepped gear shaft in one operation. For this reason, such parts should be formed in two operations. The first operation involves the forming of a hollow stepped shaft by rotary compression, while in the second operation, a gear is formed on one of the steps of the shaft.

## 1. Introduction

Hollow stepped gear shafts are more and more widely used in machine design. The high demand for hollow parts predominantly results from the fact that their strength properties are similar to those of their solid counterparts (primarily under bending and torsional loads), but at the same time, such parts are lightweight [1]. The automotive and aircraft industries are the main buyers of hollow parts. This results from the fact that—due to economic reasons and increasing fuel prices—design engineers are forced to find solutions ensuring, on the one hand, improved engine efficiency, but, on the other, lower weight of structures. In effect, fuel consumption and toxic emissions can be reduced, while the machine’s performance (power, speed, carrying capacity, maneuverability, etc.) is enhanced. This is beneficial to the end-user because it ensures lower maintenance costs [2]. Hollow parts are also increasingly popular in engineering (hollow shafts for gears, electric motors, machine tool drive elements, machine tool spindles, etc.) because reduced weight of machines and devices means lower inertia forces and, thus, higher operating parameters [3]. Hollow parts are used, among others, in various types of mechanical vehicle gear boxes, aircraft structures, machine tools, as well as in many other machines and devices that transfer the torque [4,5,6]. At present, hollow stepped gear shafts are primarily produced by machining processes, which is connected with considerable material losses amounting up to 75% [6]. Nonetheless, metal forming methods are increasingly widely used to form solid semi-finished products that are then subjected to mechanical working in order to drill holes, remove machining allowance, or form teeth [7,8,9]. This way of manufacturing hollow parts ensures reduced material consumption (compared to machining methods) and higher strength properties (material fiber continuity is retained). Forming methods for hollow parts are also more and more widely used. In this context, it is worth mentioning cold extrusion methods which are more frequently used to produce parts for the automotive industry [10,11,12]. It must, however, be emphasized that these processes must be performed in several operations due to their complexity [13,14]. This, in turn, means higher costs of both technology implementation and process realization. As a result, such processes are cost-effective only for large production batches. The efficiency of these processes can be improved by the use of unconventional manufacturing methods, enabling effective forming of hollow parts from tubes for small production batches. This approach offers two types of advantages: lower production costs resulting from lower labor costs and reduced material, energy and tool consumption, as well as improved operating parameters due to a reduction in the total weight of the machine.

Given the above, it is essential to design new manufacturing solutions that would be cost-effective even for small production batches. The development of a universal and inexpensive technique for forming hollow parts will enhance the competitiveness of companies and improve the efficiency of manufacturing processes. One of the innovative methods for producing hollow parts is rotary compression, a universal process for producing hollow machine parts. The process consists in reducing the diameter of a tube by three rotating tools; the tools are stepped rollers moving toward the axis of the workpiece. Parts produced in this way can be effectively used in aircraft structures or automotive and engineering parts, and their manufacturing is easier and cheaper than other methods [15,16,17,18]. This study proposes the use of rotary compression for producing more complex parts that have not only circular-section steps, but also gears [19,20] and worms [21]. To verify if rotary compression is an effective method of producing such parts, extensive theoretical analyses and experimental tests were conducted on forming hollow multi-stepped gear shafts. Selected results of this research are presented in this paper.

Preliminary studies revealed significant limitations of performing the rotary compression process for hollow stepped shafts with gears in one step, with the smooth (end) steps of the shaft and the central step teeth formed at the same time. It turned out that such process was very difficult to carry out due to different diameters of the formed steps, which led to differences in the tangential velocity of forming successive steps during workpiece rotation. This, in turn, led to uncontrolled slipping between the workpiece and the tools, thus causing defects in the product (slipping is particularly undesired when forming teeth). Therefore, following several unsuccessful attempts at forming stepped gear shafts in one stage, it was decided that the process would be performed in two stages. The first stage of the process involved forming smooth steps of the hollow stepped shaft (the effect of slip on product quality is less significant here), while the second stage consisted of forming teeth (the workpiece is in contact with the tools only in the gear area). In this way, it was possible to eliminate the negative effect of slip on the product quality.

## 2. Design of the Rotary Compression Process for Producing Hollow Gear Shafts

A schematic of the proposed rotary compression process for producing a hollow stepped gear shaft is shown in Figure 1. The process is performed in two operations. First, a tube is formed into a stepped shaft with smooth steps (Figure 1a), and then a gear is formed on the central step of the shaft (Figure 1b). The tools used in Operation 1 have the shape of stepped rollers. During the process, they are rotated in the same direction and move towards the axis of the workpiece. The rotating tools reduce the outside diameter of the steps on the shaft ends. Figure 2 shows the shape and geometry of the billet used in Operation 1, the semi-finished product obtained in this stage of the process, and the tools used therein.

The second operation is performed with the use of tools that have the shape of stepped rollers with a gear on the central step. The kinematics of the tools was the same as in Operation 1. Figure 3 shows the shape and geometry of the hollow gear shaft and tools. The gear on the central step of the shaft is described by the following parameters: the number of teeth *z* = 18; the normal module *m* = 2.5; the helix angle *β* = 10°, the pressure angle *α* = 20°; the helix direction—right-handed. The gear on every tool has 54 teeth with the left-handed helix angle of 10°, and its other parameters are identical to those describing the gear on the shaft. In addition, considering the process realization conditions, the tooth profile on the tools was modified in such a way as to add finishing allowance.

## 3. Numerical Analysis of the Rotary Compression Process for a Hollow Gear Shaft

To verify both the suitability of the proposed method for forming hollow gear shafts and the correctness of the employed technological and design solutions, the rotary compression process was modeled numerically by the finite element method. The proposed process was examined for failure modes such as uncontrolled slip (loss of rotation by the workpiece), workpiece, and gear defects or collapse. In addition, the relationships were investigated between individual parameters of the process and the geometry and quality of produced parts. The numerical analysis was performed with the use of the commercial simulation software package Simufact Forming v.15.

The rotary compression process was modeled in accordance with the schema shown in Figure 1 and Figure 2 (in 2 operations). Two sets of rigid tools were used in the analysis: smooth stepped rollers—set I (Figure 2) and stepped rollers with a gear on the central step—set II (Figure 3). During the process, the tools were rotated in the same direction at a constant speed, *n*, of 36 rev/min and they simultaneously moved toward the axis of the workpiece with a constant speed, *v*, of 2 mm/s (Operation 1) and 1 mm/s (Operation 2). In Operation I, the billet was a smooth tube with an outside diameter of 38 mm, a wall thickness, *t*, of 6 mm, and a length, *l*, of 100 mm. In Operation 2, the billet was a stepped shaft imported from the first stage of the analysis (without strain and temperature histories, reheat considered). Both billets were modeled as elastic-plastic materials using first-order 8-node finite elements. The billet was assigned the properties of the C45 grade steel. The material model of the C45 grade steel was taken from the material data library of the Simufact Forming software [22], and it was described by the following equation:(1)$$\sigma_{p}=1521.306\cdot e^{-0.00269\cdot T}\cdot \varepsilon^{-0.12651}\cdot e^{-0.05957/\varepsilon }\cdot {\dot{\varepsilon }}^{0.14542},$$
where *T* is the temperature (ranging 697.4–1250 °C), *ε* is the effective strain, $\dot{\varepsilon }$ is the strain rate.

The billet was preheated to 1100 °C in the first operation and to 1000 °C in the second one. The temperature of the tools in both operations was maintained constant at 100 °C. The metal-tools contact surface was described by the constant friction model with a friction factor, *m*, of 0.75 [23,24], the coefficient of heat transfer between the workpiece and tools was set equal to 20 kW/m^2^K, while the heat transfer coefficient between the workpiece and environment was set equal to 0.3 kW/m^2^K.

Figure 4 shows the FEM-modeled geometry of a semi-finished product (hollow stepped shaft) after the first operation. It can be observed that the obtained shape agrees with the required product geometry. Another observation is that the wall thickness in the region of the formed steps does not increase in a uniform manner, which primarily results from the material’s radial flow. As a result, the hole requires finishing in order to ensure a uniform wall thickness. The size of the finishing allowance is small and does not exceed 1 mm.

The numerical analysis involved determining the pattern of metal flow and variations in the workpiece temperature and wall thickness, as well as predicting fracture. Obtained results are plotted in Figure 5. In the first operation of the process, only the end and intermediate steps of the shaft are formed, whereas the central step, on which the gear is to be formed, is not subjected to reduction. This pattern of compression has a significant effect on the metal flow in the process, leading to the concentration of effective strains in the region of reduced steps (Figure 5a).

Interestingly, the highest strains are not concentrated in the region of the highest wall thickness increase (Figure 5d), but they are located at the surface layers of the steps on the shaft ends. This can be explained by the fact that additional strains are generated in a circumferential direction because of the slip between the workpiece and the tools. The forming of hollow parts entails a high risk of material cooling. This results from the much lower thermal capacity of tubes when compared to solid parts. A too high drop in temperature of a workpiece may be the main cause of defects in the cross section of reduced steps. The results demonstrate that the highest drop in the workpiece temperature occurs at the surface layers of the reduced steps (Figure 5b). The main cause of the temperature drops in these regions (to approx. 950 °C) is the contact between the workpiece and the much colder tools. The temperature in the central area of the workpiece, which has no contact with the tools, remains relatively high and is similar to the initial value (approx. 1100 °C). The observed temperature drops in the region of the formed steps (approx. 150 °C) are acceptable, and they have no effect on the stability of the forming process. A characteristic of rotary forming processes is fracture in the workpiece center (in solid parts). In the case of hollow parts, the fracture may be located at the surface of the hole. The fracture formation mechanism has not yet been exhaustively explained. One of the causes of material internal cohesion disruption may be low-cycle fatigue resulting from the cyclic stresses in the region of the hole. In the present analysis, fracture was predicted based on the Cockcroft-Latham ductile fracture criterion (2):(2)$$C=\int_{0}^{\varepsilon^{}}\frac{\sigma_{1}}{\sigma_{i}}d\varepsilon ,$$
where *σ*_1_ is the maximum principal stress, *σ_i_* is the effective stress, *ε* is the effective strain, *C* is the value of the Cockcroft–Latham integral.

The limit values of the criterion were determined in the channel-die rotary compression tests of cylindrical specimens [25,26], in which the state of stresses corresponded to that occurring in rotary forming processes. The FEM-modeled distribution of the Cockcroft–Latham ductile fracture criterion (Figure 5c) does not indicate fracture. The highest values of the Cockcroft–Latham material constant *C* located in the region of formed steps range from 0.5 to 0.7, which is several times lower than the limit value at which fracture may occur [27]. As previously mentioned, in rotary compression, the wall thickness of the formed steps is changed due to the material’s radial flow. It can be observed (Figure 5d) that apart from the regions in which the wall thickness increases, there are also regions where the wall thickness decreases. This occurs in the regions between the formed step and the non-reduced central step. However, this wall thickness decrease is insignificant and amounts to approx. 1 mm, and thus should have no effect on the product’s strength.

In a subsequent stage of the process, the gear is formed on the central step of the shaft. This process resembles the hot forming of gears with the use of three tools by deep rolling [27], the only difference being that, in the process under analysis, the billet is a hollow stepped shaft produced by rotary compression. Given the rotary compression process design, it was assumed that the semi-finished products (hollow shafts) would be cooled after Operation 1 and reheated to 1000 °C in Operation 2. For this reason, the semi-finished products from Operation 1 were imported to Operation 2 without the strain and temperature histories. A FEM-modeled shape of the gear is shown in Figure 6. It can be observed that the gear profile is correct and agrees with the required geometry. The gear formation operation is dominated by radial extrusion of the material because of the rotary and translational motion of the tools. Since in the gear region, the material is deformed to a small extent, it predominantly flows in the radial direction, opposite to the motion of the tools, and fills the spaces between the teeth on the rotating rollers. It can also be observed that the diameter of the central step is reduced, which results from the low rigidity of the hollow part. Such metal flow pattern affects the size of the initial diameter of the step on which the gear is formed. When gears are formed on solid parts, the diameter of a billet must be smaller than the reference diameter of a gear. When forming gears on hollow parts, one must allow for the cross-sectional reduction. Therefore, in the analyzed case, the diameter of the step on which the gear was to be formed was made equal to the diameter of the tooth point, decreased by the value of the module. In effect, the produced gear had the required geometry.

Figure 7 shows the FEM-modeled distributions of effective strains, temperatures, and Cockcroft–Latham ductile fracture criterion in the second operation of the process (gear formation). It can be observed that the strains are located in the vicinity of the formed gear (Figure 7a). The highest effective strains are concentrated in the bottom land of the gear, which results from the fact that the material is extruded from this region to the tooth points. The gear formation operation significantly depends on temperature that affects the deformation resistance and, hence, metal flow. The temperature data (Figure 7b) show small drops in the temperature in the region of the formed gear (by approx. 70–80 °C) resulting from the contact between the workpiece and tools. In other regions of the workpiece, where there is no contact between the material and tools, the temperature is similar to its initial values. This observation is of vital importance due to the relatively long time of gear formation. The numerical analysis also involved predicting fracture (acc. to the Cockcroft–Latham criterion) (Figure 7c). The FEM numerical results do not predict facture in the region where the gear is formed. It should, however, be emphasized that the fracture mechanism in the gear formation operation differs from that in the rotary compression of smooth steps on the shaft. In this case, the formed gear is a result of radial extrusion of the material, and the regions with the highest concentration of the ductile fracture criterion are predominantly subjected to cyclic compression. For this reason, the limit values of the Cockcroft–Latham integral should be calculated based on simple load patterns (compression or tension). In simple load patterns, the limit values of the Cockcroft–Latham integral are much smaller than those in rotary compression, and for the C45 grade steel, they are close to 1 [28,29].

## 4. Experimental Tests of the Rotary Compression of Stepped Gear Shafts

Experimental tests of rotary compression were performed on a forging machine designed and constructed at the Department of Computer Modeling and Metal Forming Technologies at the Lublin University of Technology (Figure 8). The machine is made of segments and consists of: a support frame—1, a drive system—2, a pinion stand—3, a roll stand—4, a hydraulic drive system for the forming rollers—5, a control and power supply system—6, a measuring system (Figure 8b). The two-operation rotary compression process was performed in the roll stand with three radially moving slides equipped with bearing-mounted working shafts. The translational motion of the slides was synchronized to ensure uniform displacement of the tools during the forming process.

Force and kinematic parameters of the rotary compression process were measured with the use of specifically constructed measuring system. The system consisted of a displacement transducer with limit switch for monitoring the current speed and location of the tools, torque transducer for digital measurement of torque during the forming process, and pressure transducers for measuring pressure in the servo-motors and determining the tool load on the workpiece. Data measured by the transducers were stored in the computer with a specifically designed data recording system (Figure 8b), its main element being a National Instruments USB-6008 device (National Instruments Corp., N Mopac Expwy, Austin, TX, USA). The data recording system was managed by a special LabVIEW (ver. 2016, National Instruments Corp., N Mopac Expwy, Austin, TX, USA).—designed application for digital measurement of transducer-recorded signals.

The experiments were performed using two sets of tools: smooth rollers (Figure 9a) and rollers with gears (Figure 9b). The shape of the tools was the same as that used in the numerical analysis. Other parameters of the process were also identical to those applied in the FEM modeling. The experimental tests were performed using C45 steel sleeves, the dimensions of which are shown in Figure 2a. The material in as-received condition was normalized and according to the certificate had the following properties: tensile strength Rm = 560–640 MPa, yield point Re = 275–350 MPa, elongation A = 14–17%. The sections of tubes used in Operation 1 were first preheated to approx. 1100 °C in an electric chamber furnace and then fed into the working space of the machine (created by three rotating rollers) using tongs. The tools were rotated in the same direction at the constant speed *n* = 36 rev/min and they simultaneously moved radially with the constant speed *v* = 2 mm/s. The tools made the billet rotate and reduced the diameter of the steps on the shaft ends. Once the slides traveled the path length corresponding to the required diameter reduction, their translational motion was stopped to begin sizing the shape of the workpiece in successive revolutions of the rollers. In the final stage of the process, the tools move apart in a radial manner, and the semi-finished product (Figure 10a) is removed from the working space of the machine and subjected to slow air cooling.

In the second operation of the process, the working shafts of the machine were equipped with a set of toothed rollers for forming a gear on the central step of the shaft obtained from the first operation of the process (Figure 10b). Prior to Operation 2, the tube was first cleaned of scale and then preheated to 1000 °C. After that, it was mounted between the tools which were rotated in the same direction at 36 rev/min and were moving toward the axis of the workpiece with the speed *v* of 1 mm/s.

A shape of the semi-finished product after Operation 1 and that of the finished part are shown in Figure 11. The experimental product shape and dimensions show good agreement with the theoretical assumptions (the designed product shape) and numerical results. The surface of the semi-finished products after the first operation is smooth and free from defects. No internal defects such as fracture or overlap can be observed. In a likewise manner as in the FEM simulations, the workpiece wall thickness in the region of the reduced steps increased in a non-uniform manner and cavities were formed on the end face of the steps on the shaft ends. The gear formed in the second operation of the process shows a relatively high agreement with the design assumptions and FEM results. Naturally, the accuracy of the gear formed in this way is relatively low (because of the process conditions), which consequently requires adding the finishing allowance. Nevertheless, the results demonstrate that the proposed process is an effective method for forming hollow stepped gear shafts.

The teeth are formed with small material allowance that will be removed in finishing (Figure 12). It is worth stressing the fact that even though the process is conducted under hot working conditions, the deviations are small and do not exceed 0.2 mm, which can be considered a satisfactory result.

The study also involved analyzing the force parameters (torques and forces) that affect the practical realization of the process. The plots in Figure 13 show high agreement between the experimental findings and FEM results in terms of both quality and quantity. The small discrepancies between the torque results may be caused by the difficulty in accurate modeling of thermal conditions and material properties of the analyzed steel.

## 5. Summary and Conclusions

The FEM analysis results of the rotary compression process have confirmed that this method can be used for forming hollow stepped gear shafts. The proposed method for forming hollow parts is innovative and has not yet been employed in industrial production. Products of this type are predominantly produced by machining methods, which is connected with high material and energy consumption as well as high labor costs. By forming such parts from tubes, it is possible to significantly reduce the material consumption and the structure’s weight, while at the same time, to improve the strength properties of the structure. As a result, the operating costs of machines and devices can be considerably reduced. Owing to its many advantages, the proposed method can be used both in small batch production, e.g., in the aircraft industry (stepped axles and shafts made of non-ferrous metal alloys), and in mass production (automotive, engineering). Unfortunately, however, previous studies demonstrated that it is impossible to form stepped gear shafts in one operation. This primarily results from considerable differences between the tangential velocities of the steps and gear, and may cause defects in the smooth steps and the gear on the shaft, thus leading to product failure.
The results of this study lead to the following conclusions.It is characteristic of hollow shafts formed by rotary compression that the wall thickness of reduced steps is increased.The diameter of the step on which a gear is to be formed should allow for the cross-sectional reduction.During the second operation of the rotary compression process, the internal wall of the workpiece is deformed to a small extent, which facilitates the formation of the gear.

## Figures and Tables

**Figure 1 materials-13-05718-f001:**
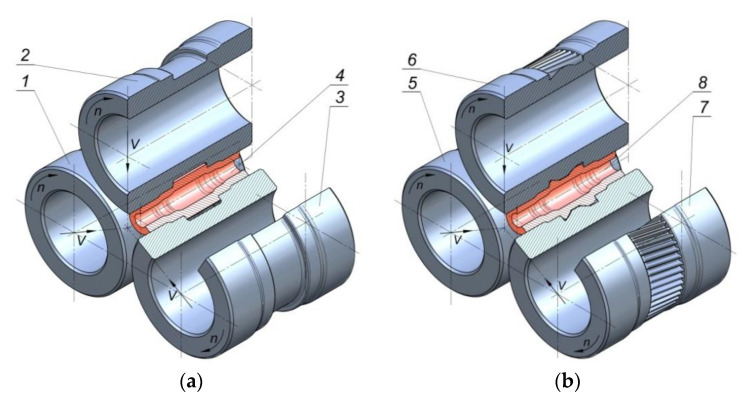
Schematic design of the rotary compression of a hollow stepped gear shaft: (**a**) Operation 1—formation of a shaft with smooth steps, (**b**) Operation 2—formation of a gear; 1, 2, 3–tools used in Operation 1, 4—stepped shaft, 5, 6, 7—tools used in Operation 2, 8—hollow stepped shaft with a gear.

**Figure 2 materials-13-05718-f002:**
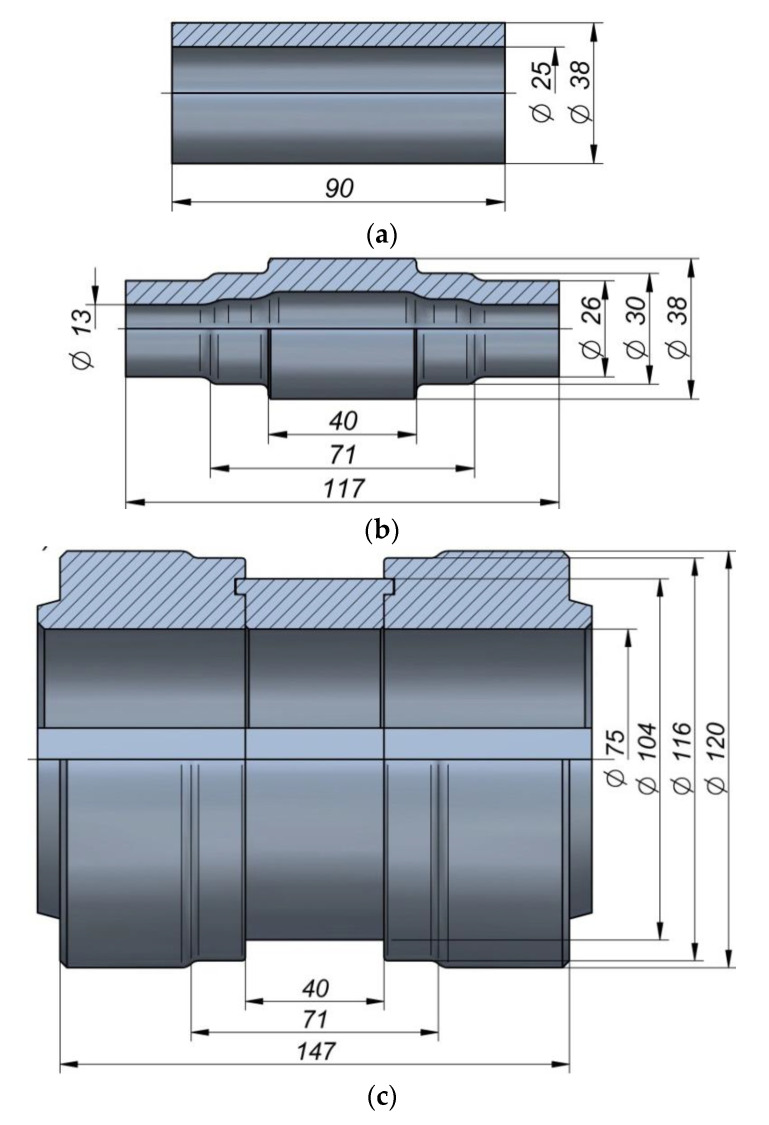
Shape and dimensions of: billet (tube)(**a**), semi-finished product (**b**), tools (**c**), in Operation 1.

**Figure 3 materials-13-05718-f003:**
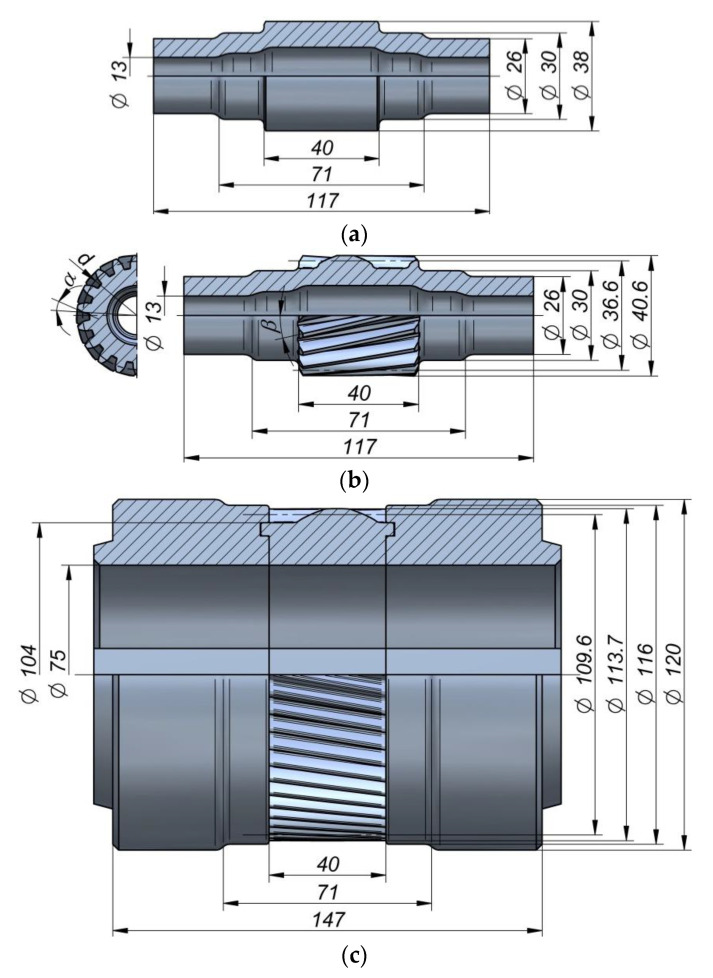
Shape and overall dimensions of: semi-finished product (**a**), product (**b**), tools (**c**), in Operation 2.

**Figure 4 materials-13-05718-f004:**
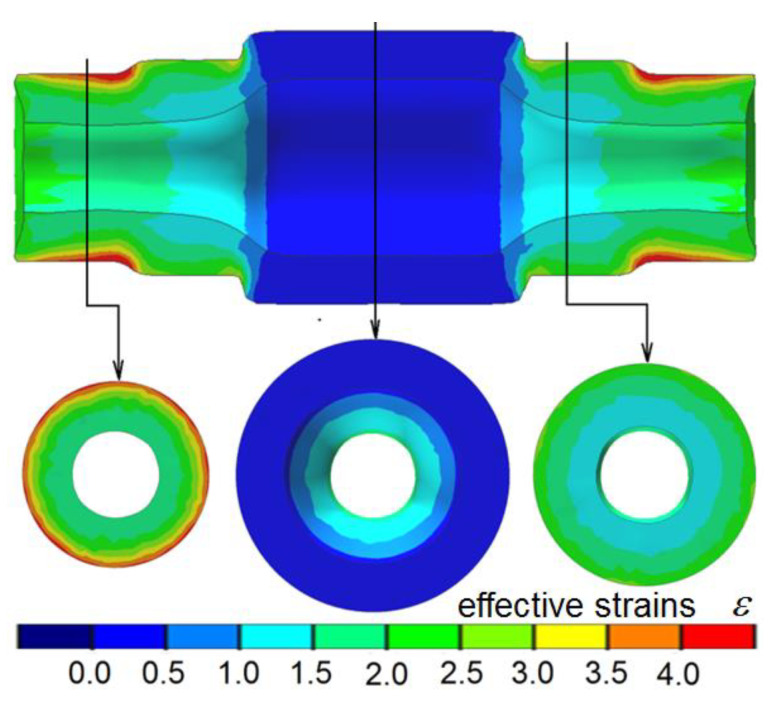
FEM-modeled shape of a stepped gear shaft after Operation 1 and the distribution of effective strains.

**Figure 5 materials-13-05718-f005:**
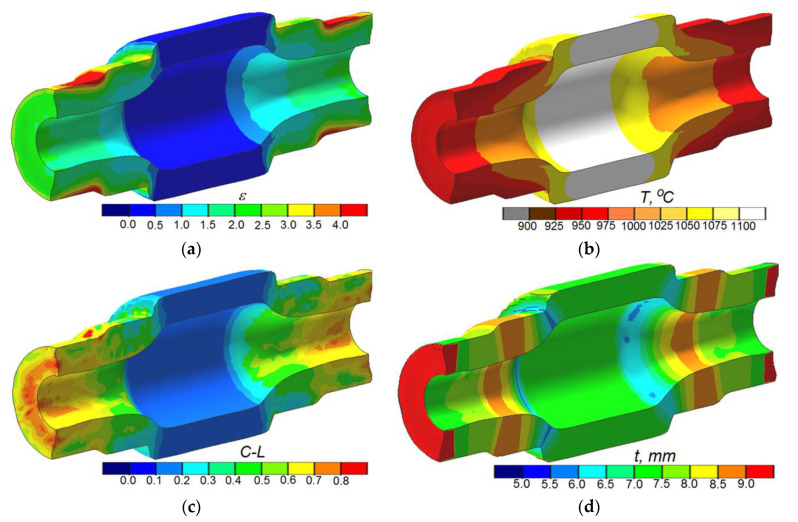
FEM-modeled distributions of: (**a**) effective strain, (**b**) temperature, (**c**) Cockcroft–Latham ductile fracture criterion, (**d**) wall thickness at the end of Operation 1.

**Figure 6 materials-13-05718-f006:**
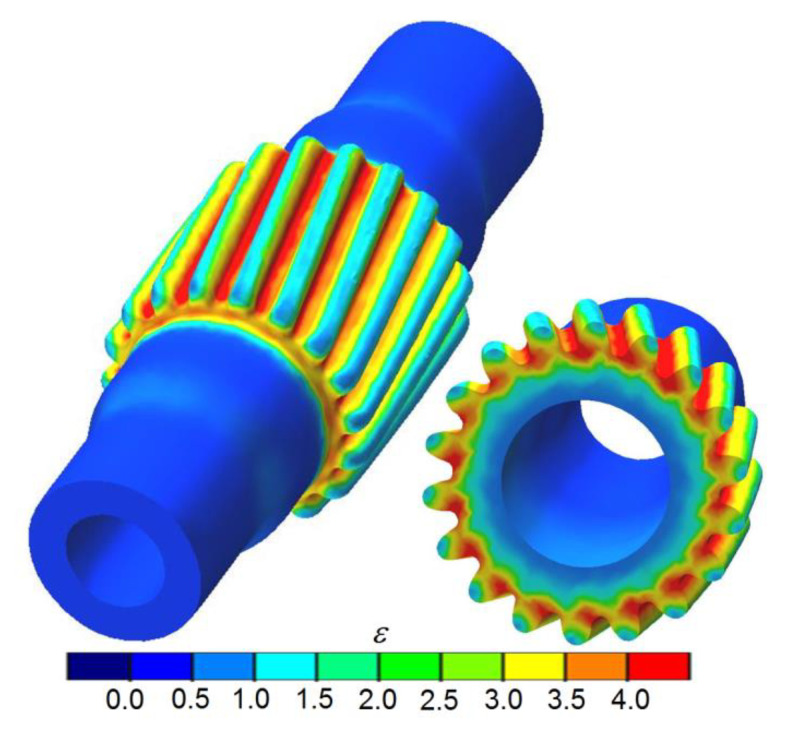
FEM-modeled shape of the gear formed on the shaft and the distribution of effective strains.

**Figure 7 materials-13-05718-f007:**
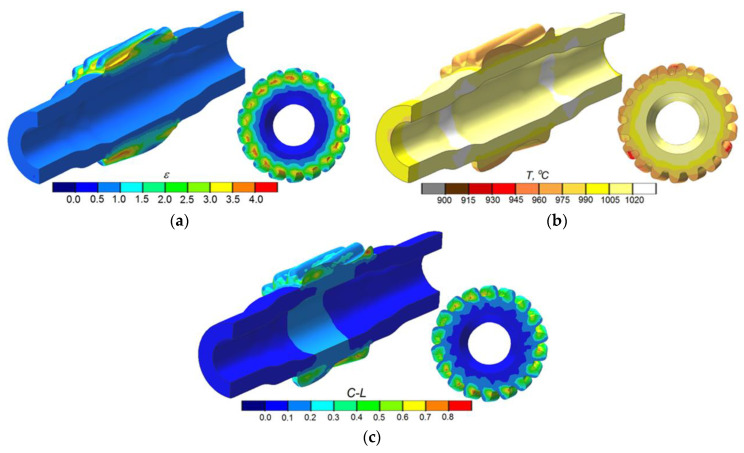
FEM results in the second operation of the rotary compression of a hollow stepped gear shaft: (**a**) effective strain, (**b**) temperature, (**c**) Cockcroft–Latham ductile fracture criterion.

**Figure 8 materials-13-05718-f008:**
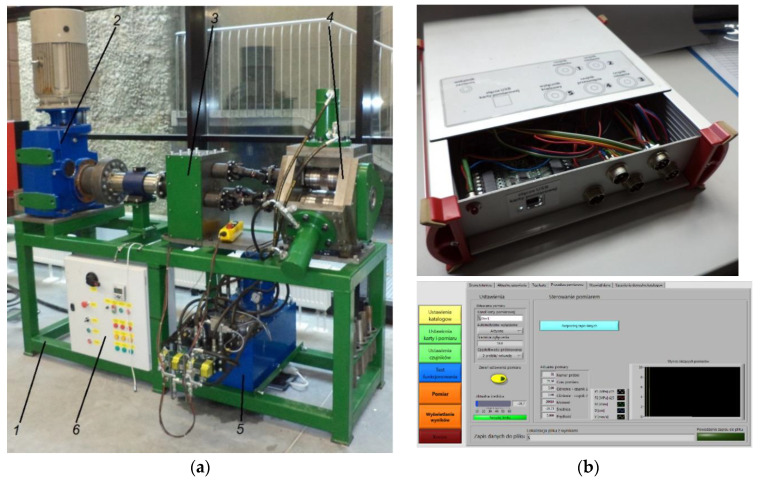
Rotary compression machine for hollow parts: (**a**) view of the machine, (**b**) measuring system with user interface.

**Figure 9 materials-13-05718-f009:**
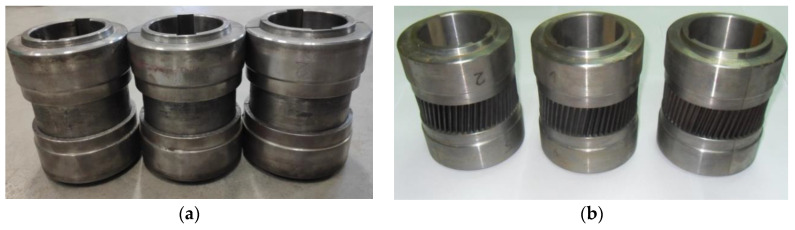
Tools used in the rotary compression of hollow gear shafts: (**a**) rollers with smooth steps used in Operation 1, (**b**) rollers with gears on the central step used in Operation 2.

**Figure 10 materials-13-05718-f010:**
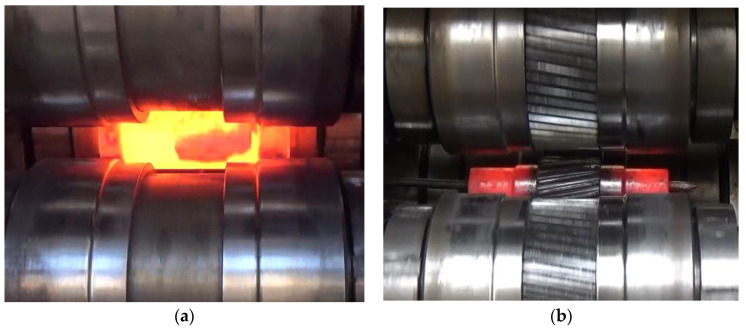
Rotary compression of a hollow stepped gear shaft: (**a**) rotary compression of a hollow shaft with smooth steps, (**b**) formation of a gear on the central step of the shaft.

**Figure 11 materials-13-05718-f011:**
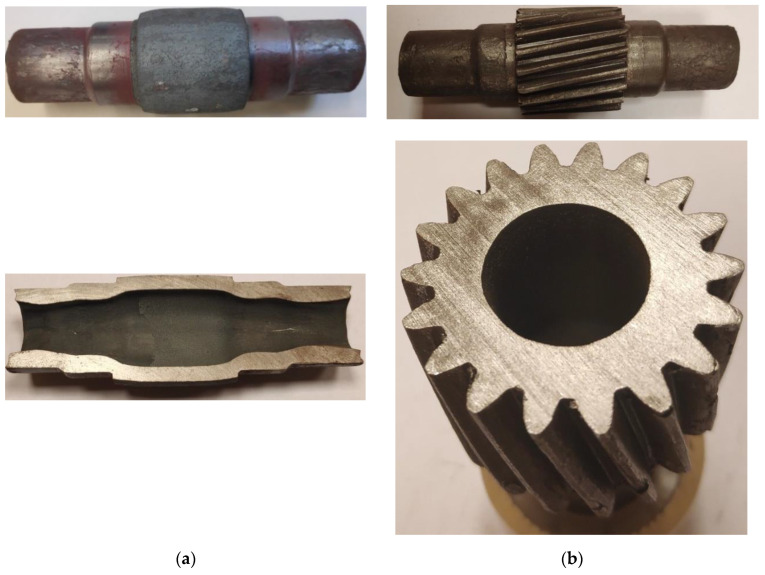
Hollow shafts with smooth steps formed in the first operation of rotary compression (**a**) and gear formed in the second operation of rotary compression (**b**).

**Figure 12 materials-13-05718-f012:**
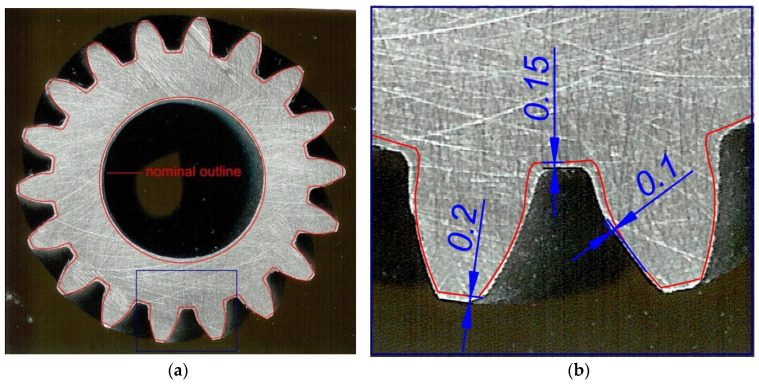
Experimental and FEM tooth profiles with material allowance: (**a**) cross section, (**b**) deviation outline teeth.

**Figure 13 materials-13-05718-f013:**
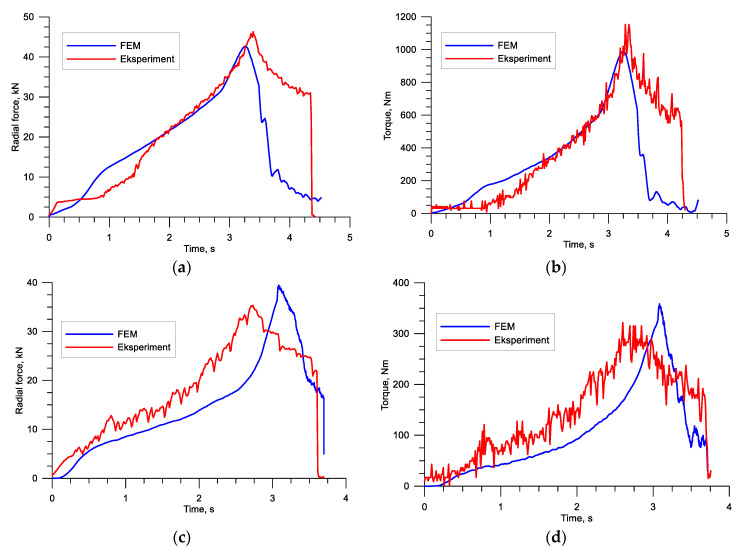
Force parameters in the rotary compression of a hollow stepped gear shaft: (**a**) radial force (Operation 1), (**b**) torque (Operation 1), (**c**) radial force (Operation 2), (**d**) torque (Operation 2).

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
