# Peer review of "A Rotary Compression Process for Producing Hollow Gear Shafts"

_materials, 2020, doi:10.3390/ma13245718_

Round 1

Reviewer 1 Report

This manuscript is well written with interest subject. However, the authors should supplement the following issues in the manuscript.

  1. To help readers understand, it is necessary to schematically express where the pressure angle (α), and helix angle (β) mean together with the size of the gear.
  2. Built-in material properties of C45 grade steel were used in the FEM analysis. Did you verify by actually testing material properties?
  3. In figure 4, it should be indicated that the legend means effective strains.
  4. In the FEM analysis, it was mentioned that the temperature of the tool is fixed and maintained at 100ºC, but in the actual process, the temperature of the tool rises when it comes into contact with the workpiece preheated to 1000~1100ºC. Is there any evidence on whether the results can be trusted even if the analysis proceeds ignoring theses phenomena?
  5. The manuscript has only explained the advantage of 2-steps process, utilizing FEM analysis and experimental results when forming a gear. In order to help readers understand, what kind of problems arise when gears are formed with 1-step process, so it is good to show the advantages of 2-steps process together with FEM analysis and experiment results.
  6. As mentioned in the manuscript, if the dimension of the gear manufactured by 2-steps process fits well with the expected result, it is necessary to compare it with a quantitative value and show it using figure.
  7. In the FEM analysis, was the residual stress occurring after step 1 considered in step 2?

Author Response

The authors wish to thank the Reviewer for a thorough review and all valuable comments.The authors wish to thank the Reviewer for a thorough review and all valuable comments.

Detailed responses to comments are attached.

Reviewer 2 Report

This paper presents a two-step forming process for producing hollow gear shaft. Finite Element simulation and experiments were used to perform the study.

Some issues and recommendations are mentioned below.

1. This paper presents an elementary manufacturing process, and the innovation in research is missing. In its current form, the manuscript seems more like a technical report than a scientific paper. To increase the scientific level of the manuscript, I suggest including an experimental study on the evolution of material properties (for example the microhardness) and microstructure during the forming process.

2. What kind of material was employed in the experiments? In addition, it is not suitable to present the variation of the experimental force without presenting the mechanical properties of the material.

3. In the Finite Element simulation of the forming process, the material properties were taken from the material library of the Simufact Forming program. As it is known, the properties of a material can vary from manufacturer to manufacturer and even from batch to batch. How do the authors explain that the numerical results approximate the experimental data very well?

Author Response

We wish to thank the Reviewer for an in-depth review of our manuscript and all valuable comments.

Detailed responses to comments are attached.

Round 2

Reviewer 2 Report

The authors have reasonably answered to all my comments and made the necessary changes.